# Fatigue Characteristics of 7050-T7451 Aluminum Alloy Friction Stir Welding Joints and the Stress Ratio Effect

**DOI:** 10.3390/ma15228010

**Published:** 2022-11-13

**Authors:** Hanji Zhu, Giuseppe Lacidogna, Caiyan Deng, Baoming Gong, Fei Liu

**Affiliations:** 1Key Laboratory of Advanced Joining Technology of Tianjin, Department of Materials Science and Engineering, Tianjin University, Road Weijin 92, Tianjin 300072, China; 2Department of Structural, Geotechnical and Building Engineering, Politecnico di Torino, Corso Duca degli Abruzzi 24, 10129 Torino, Italy; 3China Jingye Engineering Technology Co., Ltd., Xitucheng Road 33, Beijing 100088, China

**Keywords:** fatigue, 7050-T7451 aluminum alloy, FSW, stress ratio, coarse inclusion phase

## Abstract

The fatigue crack initiation and growth characteristics in 7050-T7451 aluminum alloy butt joints subjected to different stress ratios and owing to friction stir welding (FSW) were investigated using fatigue tests for stress ratios of 0.1, 0.3, and 0.5. The difference between the fatigue crack initiation in the base material (BM) and FSW joints, related to coarse secondary phases, was explored using scanning electron microscopy (SEM). Accordingly, Al_23_CuFe_4_, Al_7_Cu_2_Fe, and Al_2_Mg_3_Zn_3_ were the preferred joint crack initiation locations, whereas Mg_2_Si was the major fracture initiation point of the parent material, and cracks tended to propagate along dense, coarse secondary phases, becoming more pronounced for larger cracks. In addition, as the stress ratio increased, non-Mg_2_Si phase fracture initiation points appeared in the BM. Meanwhile, the quantity of non-Mg_2_Si phases in the joints continued to increase, and the crack initiation sites became increasingly concentrated in the TMAZ-HAZ region.

## 1. Introduction

High-strength aluminum alloys have been widely used in the aerospace industry and high-speed vehicles owing to their low density and high strength-to-weight ratio [1]. However, owing to their high thermal conductivity and strong oxidation susceptibility, 7××× aluminum alloys have been difficult to be jointed using conventional fusion-welding techniques. These issues were resolved using the friction stir welding (FSW) technology, which has led to the widespread use of poorly welded high-strength aluminum alloys [2,3,4]. Therefore, the fatigue performance of high-strength aluminum alloys and their welded joints is a crucial factor affecting the safety and reliability of structures subjected to asymmetric alternating loads during their service time [5,6].

Recent studies have shown that coarse secondary phases play an important role in the fatigue behavior of aluminum alloys. Zheng et al. [7] investigated the effects of secondary-phase particles on the fatigue behavior of 2524-T34 aluminum alloy sheets and discovered that fatigue cracks primarily occurred at these secondary-phase particles and secondary-phase particle/substrate contact interfaces. According to Payne et al. [8], secondary phases are almost exclusively responsible for fatigue cracking, with most fatigue cracks starting at the loci of iron-rich secondary phases. Harlow et al. [9] found that only 3.5% of iron-rich secondary-phase cracks were subjected to cyclic loading, whereas approximately 87% of iron-rich secondary phases were observed in high-stress zones. Using nanoindentation techniques, Sudhanshu et al. [10] quantitatively and qualitatively evaluated secondary phases and discovered that the elastic moduli of iron-rich secondary phases were in the 130–169 GPa range, while those of Si-containing secondary phases were in the 44–97 GPa range. Using a four-point bending test on the AA7075-T651 alloy, Jin et al. [11] discovered that the pre-cracked iron-rich secondary phase was responsible for the initiation of fatigue cracks. Another popular subject in fatigue research is the effect of mean stress. Deng et al. [12] performed ultrasonic fatigue tests on the 7050-T7451 aluminum alloy’s FSW joints at R = −1, −0.3, and 0.3 and discovered that cracks primarily originated at the surface for higher mean stress levels, whereas for lower mean stress values, cracks were competitively nucleated in superficial and internal regions. Ma et al. [13] investigated the stress ratio effects on the fatigue-crack growth characteristics of 5083 aluminum alloys, and reported that the fatigue-crack propagation rate increased with increasing stress ratio, whereas the propagation threshold and fracture toughness decreased. Moreover, the average stress relaxation phenomenon was observed in the base material (BM) and weld regions as cyclic deformation proceeded. This phenomenon was more pronounced in the early stages of fatigue and weakened with increasing propagation, according to a thorough and in-depth study on the cyclic deformation of Ti-6Al-4V and Ti17 BMs and joints [14]. Jata et al. [15] studied the effect of the microstructure on the fatigue properties of the 7050-T7451 aluminum alloy’s FSW joints and reported that the crack propagation rate in the stir zone (SZ) decreased at R = 0.33, while that in the heat-affected zone (HAZ) increased. At R = 0.7, the difference between the crack propagation rates for different zones was significantly smaller, owing to the intergranular damage mechanism in the SZ and residual stress in the HAZ. Moreover, the failure location in the studied FSW joints correlated well with the microstructure heterogeneity. Besel et al. [16] discovered that the SZ and the thermal–mechanical affected zone (TMAZ) of the 5024 aluminum alloy’s FSW joint line tended to failure. The fatigue performance of the 6N01 aluminum alloy’s FSW joints was studied by Sillapasa et al. [17], who reported that fatigue-related failures correlated with the hardness distribution in different joint regions and were more likely to occur in the HAZ with the lowest hardness. The strength gradient in the weld caused the plastic to concentrate in the HAZ, according to White et al. [18], who studied the impact of pre-stretching on the fatigue fracture initiation and mechanical behavior of the AA7050 aluminum alloy’s FSW joints. However, the role of the coarse secondary phase has not yet been fully determined. The relationship between the fatigue behavior and coarse secondary-phase particles in FSW joints remains elusive.

As demonstrated above, while there have been numerous studies on secondary phases of aluminum alloys, the mechanism of crack initiation and the effect of the stress ratio of the different secondary phases in the base material and joints are not well studied in a systematic manner.

In this study, high-cycle fatigue tests were conducted on the aluminum alloy 7050-T7451 and its FSW joints, subjected to the stress ratios of 0.1, 0.3, and 0.5. The relationship between the coarse secondary phases and the preferred crack-initiation sites is discussed, and the irregular distribution of crack-initiation locations in joints subjected to various stress ratios is investigated.

## 2. Materials and Methods

### 2.1. Materials and FSW Joints

The BM was a rolled plate of the aluminum alloy 7050-T7451 with the dimensions of 12 mm × 1500 mm × 3000 mm. The chemical compositions and mechanical properties are listed in Table 1 and Table 2, respectively. The rolled plate was divided into two flat plates with the dimensions of 700 mm × 70 mm × 12 mm, and the welding procedure was carried out using an HT-JM16 gantry FSW machine supplied by Aerospace Engineering Equipment (Suzhou, China) Co., Ltd. The stirring head was a threaded cone, and an I-shaped groove was adopted, with the welding direction parallel to the rolling direction of the plate. The specific welding process parameters are listed in Table 3 [19], while the friction-stir processing is shown schematically in Figure 1.

### 2.2. Hardness Test and Characteristic Areas

A hardness test across the entire weld joint was conducted on the middle line of the thickness, using an HV-1000A microhardness tester (made in Shanghai, China). The hardness test utilized an indent spacing of 0.15 mm for the TMAZ and SZ and 0.2 mm for the BM and HAZ, a load of 3 kgf, and a loading and holding time of 15 s. The specimens used for the hardness test were ground using a wet sandpaper to 7000 mesh before being polished with a diamond sus-pension (particle size, 0.25 μm) to remove the influence of the surface-hardening layer. Figure 2a shows the W-shaped hardness distribution for the FSW joints. Accordingly, an FSW joint could be divided into four areas: (1) BM, (2) HAZ, (3) TMAZ, and (4) SZ. The advancing side was the tangential direction of the stirring needle rotation, while the other side was the retreating side (RS), as shown in Figure 2b.

### 2.3. Fatigue Test

Fatigue tests on specimens subjected to different stress ratios were conducted using a GPS200 high-frequency fatigue test machine in an ambient environment. The specimens of the 7075-T7451 aluminum alloy BM and FSW butt joints in Figure 3 were tested until failure at the three stress ratios of R = 0.1, 0.3, and 0.5 at the frequency of approximately 100 Hz. The specimen mark was in the ‘type/stress/ratio number’ format. For instance, serial number M-1-1 designated the first fatigue specimen of the BM with the stress ratio of 0.1, while J-3-3 designated the third fatigue specimen of the FSW joint with the stress ratio of 0.3.

A comparison of the fatigue S–N curves for various stress ratios of the BM and joints is shown in Figure 4. At 10^7^ cycles, the fatigue strengths for the BM were 254, 228, and 175 MPa, respectively, whereas for the FSW joints, the conditional fatigue strengths were 193, 177, and 138 MPa, respectively. In addition, the fatigue strengths of the FSW joints were approximately 20% lower than those of the BMs subjected to the same stress ratios; in all the cases, the fatigue strengths decreased as the stress ratio increased. Test results are given in Appendix A.

### 2.4. Metallography Preparation and Analysis

The specimens for the optical microscopy (OM) analysis were cut perpendicular to the welding direction and prepared following the metallography procedures described in Section 2.2. Keller’s reagent was used for the etching after polishing. Metallographic observations were performed using an OLYMPUS-GX51 microscope (Shinjuku, Japan). The fracture surface was observed using a JSM-7800F scanning electron microscope (SEM, JEOL, Akishima, Japan) equipped with an energy spectrometer and a backscatter probe. By selecting appropriate gray threshold values, secondary phases were identified and analyzed using the Image-Pro Plus 6.0 software.

## 3. Results

### 3.1. Microstructure Observations

#### 3.1.1. Metallographic Analysis

The welding process can have a significant impact on the microstructure, and large angular grain boundaries are dangerous locations for fatigue failure of the specimen [20]. The microstructures of the characteristic regions of the FSW joints are shown in Figure 5. Fine and uniformly distributed equiaxial grains were generated during the fully recrystallized process in the SZ, owing to the severe deformation and sufficient frictional heat, as shown in Figure 5a. The TMAZ between the HAZ and SZ was characterized by a highly deformed structure, whereas the lower thermal input in this zone did not provide sufficient driving force for dynamic recrystallization. Because the HAZ was unaffected by stirring, the grain structure was virtually identical to that of the BM, as shown in Figure 5b. The black particles in Figure 5 are the coarse secondary phases in the matrix, which exhibit different morphologies owing to the thermal–mechanical variation of the welding process. In contrast to the TMAZ, where only a slightly coarse secondary phase was observed owing to the lower temperature, the secondary-phase size in the SZ was much finer after directly contacting the stirring pin, as shown in Figure 5c. The same coarsening phenomenon took place in the HAZ, although the comparison between Figure 5d,e indicates that the size of the coarse secondary phase in the HAZ was larger than that in the SZ. As shown in Figure 5f, the BM contained two different types of coarse secondary-phase particles, that is, white and gray–black particles, as reported by SEM.

#### 3.1.2. Coarse Secondary Phases

The coarse secondary phases were characterized via selected-area diffraction by TEM analysis, and the results are shown in Figure 6. The gray–black coarse secondary phase in Figure 5f is Mg_2_Si, while the white coarse secondary phases mainly consist of Al_23_CuFe_4_, Al_7_Cu_2_Fe, and Al_2_Mg_3_Zn_3_, based on the atomic number calibration analysis. In the BM, only the Mg_2_Si phase appears to be gray–black, while all the other non-Mg_2_Si phases are white. The coarse secondary phase in the BM is categorized into Mg_2_Si and non-Mg_2_Si phases, in agreement with the gray difference of the coarse secondary phase under the backscatter lens, to enable further statistical analysis.

### 3.2. Fatigue Crack-Initiation Mechanism

The mechanism of the fatigue crack emergence in the high-circumference range was the primary focus of this study. Because fatigue crack initiation and early growth account for most of the fatigue life, the impact of the FSW on the microstructure complicates the fatigue crack-initiation mechanism. Fine precipitates are common sites for fatigue crack nucleation in aluminum alloys. The constituent particle’s size and shape are important characteristics that influence crack nucleation. Additionally, differences in stiffness and thermal expansion coefficients between the inclusion and the surrounding matrix can introduce a localized stress concentration in and around a particle which increases the likelihood of fatigue crack formation. Fatigue cracks often form at inclusions by one of three mechanisms: inclusion cracking, debonding of the interface between the inclusion and matrix, or cracking at lines of slip in the surrounding matrix [21]. According to experimental observations, two distinct crack-nucleation mechanisms can be identified: (i) cracks nucleate in the bulk of the coarse secondary phase, and (ii) cracks emerge at the interface between the secondary phase and the matrix. It is widely thought that the former mechanism dominates crack nucleation in the case of non-Mg_2_Si phases, while both mechanisms coexist in the case of Mg_2_Si secondary phases. 

Figure 7a,b show that the initiation, stable propagation, and final fracture regions can be identified. Furthermore, as shown in Figure 7c,d, some coarse secondary phases on the surfaces of the BM and FSW joint specimens could be identified as sole crack-initiation sites. According to the studies of energy spectra, the coarse secondary phases at the crack initiation sites were Mg_2_Si for the BM and Al_23_CuFe_4_ for the joints (see Appendix B for the morphology description). Moreover, the statistical results of the fatigue fractures in Figure 8 demonstrated that 78 of 97 fatigue fractures in the BM and joints were coarse secondary-phase-initiated cracks, while 32 of 38 of the BM specimens were cracked by the coarse secondary-phase Mg_2_Si, whereas the non-Mg_2_Si phase (Al_2_Mg_3_Zn_3_, Al_23_CuFe_4_, and Al_7_Cu_2_Fe) cracking accounted for 32 of the 40 fractures in the joint specimens. The gray–black Mg_2_Si phase was the primary cracking phase for the BM specimens, whereas the white secondary phase was the primary cracking site for the joint specimens. Therefore, secondary phases are the predominant fatigue crack-initiation sites in the 7050-T7451 aluminum alloy, regardless of the BM and FSW joints.

The major cracking phase of the BM specimens was Mg_2_Si, which was the only stable intermetallic phase in the Mg-Si binary alloy. Although the Mg_2_Si phase has excellent mechanical properties, it is highly brittle and has low ductility, as shown in Figure 9 [22]. The backscattered electron images in Figure 10 illustrate the status of the coarse secondary-phase Mg_2_Si on the fracture surface, which is fragmented and debonded from the matrix. Figure 10b shows the small craters owing to the Mg_2_Si debonding from the substrate and the serrated cracks at the edges, which indicates that cracks tended to initiate in the bulk Mg_2_Si, with the subsequent coalescence and deflection of microcracks along the interface leading to serrated cracks. Although the Si content in the 7050 aluminum alloy was only 0.10%, its size was similar to that of Al_2_CuMg, and most of the Mg_2_Si phase observed at the broad transverse side was internally broken. However, the non-Mg_2_Si phase was tightly bound to the matrix, as shown in Figure 9b, causing an interfacial stress concentration or singularity owing to the elastic deformation inconsistency. This type of crack nucleation exhibited distinct fatigue cracks and extension marks on the bond surface on the microscopic level, as shown in Figure 11.

Figure 12a,b show the craters generated following the debonding of the Mg_2_Si phase, which indicates that the crack initiation mechanism of the Mg_2_Si phase in the joints was the same as in the BM. The thermomechanical deformation of FSW might have worsened the bonding strength between the Mg_2_Si phase and the matrix. Figure 13a shows the internal cracking of the large non-Mg_2_Si phase (50 μm), which was first subjected to external forces. Subsequently, the cracks gradually spread across the interface into the matrix as a result of the cyclic loading. As shown in Figure 13b, the crack growth in the non-Mg_2_Si phase can be divided into two distinct regions: (1) a smooth region created by instantaneous fracture close to the specimen surface and (2) a rough region with signs of stable crack extension far away from the specimen surface.

The distribution of the coarse secondary phase on the fracture surface, for various phases of the fatigue crack extension, was investigated for determining the impact of the coarse secondary phase on the fatigue crack extension. The results are shown in Figure 14. The Image Pro J 6.0 software was used for determining the geometrical parameters of the secondary phases. The processing results are shown in Figure 14d–f,j–l. According to the distribution of the coarse secondary phase on the fracture surface, the density of the phase gradually increased as the crack propagated. When the crack propagates in the full joint specimen, it is more likely to propagate along the direction where the strength of the entire joint is the lowest and the coarse secondary phase is the most densely distributed. This is also confirmed in the article by Liu [23]. This provided credible experimental results for future research.

### 3.3. Effect of the Stress Ratio on the Fatigue Behavior

The numbers of the cracking phases in the joint and BM specimens for the Mg_2_Si and non-Mg_2_Si phases for the three stress ratios are listed in Table 4. For the BM specimens, the primary cracking phase was Mg_2_Si. However, as the stress ratio increased, non-Mg_2_Si phase cracks appeared. For the joint specimens, most of the cracked phase was the non-Mg_2_Si phase at low mean stress values, and this phenomenon became more pronounced as the stress ratio increased. In the BM, the stress concentration at the non-Mg_2_Si and matrix interface was insufficient to cause crack initiation when the stress was relatively low. For the FSW joints, the specimens were primarily broken in the non-Mg_2_Si phase, and internal cracking was more likely to extend to the matrix. This behavior became more obvious at higher stress ratios. Figure 15 summarizes the average areas of the crack initiation-related secondary phases for the different specimens subjected to the different considered stress ratios. At the stress ratios of 0.1, 0.3, and 0.5, the average areas of the cracked phases for the BM specimens were 514 μm^2^, 432 μm^2^, and 378 μm^2^, and the average areas of the cracked phases for the joint specimens were 323 μm^2^, 293 μm^2^, and 288 μm^2^, respectively. The cracking phase areas for the BMs were greater than for the joint specimens and decreased noticeably as the stress ratio increased.

### 3.4. Effect of the Stress Ratio on the Fracture Location in Joint Specimens

The fracture-location statistical analysis was performed for all specimens, and only the results for the FSW specimens are plotted in Figure 16 for clarity. It was found that the fracture locations in the BM specimens were randomly distributed along the gauge; in contrast, only a few FSW specimens were fractured at the SZ or HAZ, whereas most failures occurred preferentially in the TMAZ-HAZ. It was also found that as the stress ratio increased from R = 0.1 to R = 0.5, the crack initiation sites tended to aggregate in the TMAZ and HAZ regions.

## 4. Conclusions

This study determined that coarse secondary phases served as the main fatigue crack-initiation sites for the BM and FSW joints of the 7050-T7451 aluminum alloy. Fractographic analyses demonstrated that Al_23_CuFe_4_, Al_7_Cu2Fe, and Al_2_Mg_3_Zn_3_ were the preferred joint crack-initiation sites, whereas Mg_2_Si was the major fracture-initiation point of the parent material. Two distinct crack-nucleation mechanisms were identified: (1) crack nucleation in the bulk of the non-Mg_2_Si phases such as Al_23_CuFe_4_, Al_7_Cu_2_Fe, and Al_2_Mg_3_Zn_3_ and (2) crack initiation at the Mg_2_Si–matrix interface. The former mechanism accounted for 77.6% of the FSW joint specimens, whereas 83.3% of the cracks formed in the BM specimens owing to the latter mechanism. In addition, as the stress ratio increased, non-Mg_2_Si phase fracture initiation points appeared in the BM. Meanwhile, the number of non-Mg_2_Si phases in the joints continued to increase, and the crack initiation sites became increasingly concentrated in the TMAZ-HAZ region.

## Figures and Tables

**Figure 1 materials-15-08010-f001:**
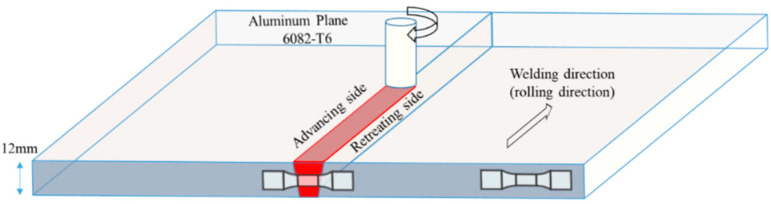
Schematic of the friction-stir processing.

**Figure 2 materials-15-08010-f002:**
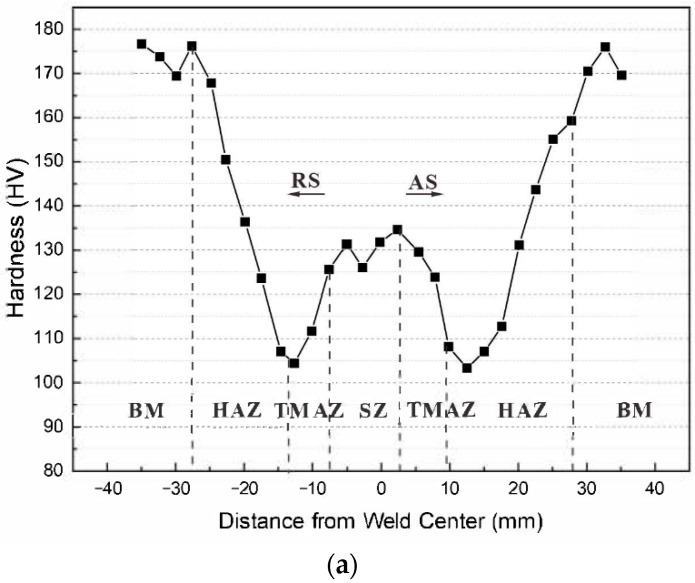
Results of the hardness test: (**a**) hardness distribution in FSW joints and (**b**) macroscopic morphology of aluminum alloy FSW joints.

**Figure 3 materials-15-08010-f003:**
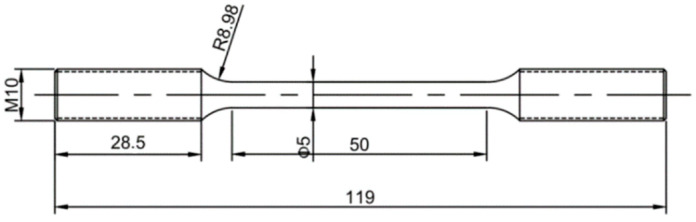
Specimen dimensions for fatigue studies.

**Figure 4 materials-15-08010-f004:**
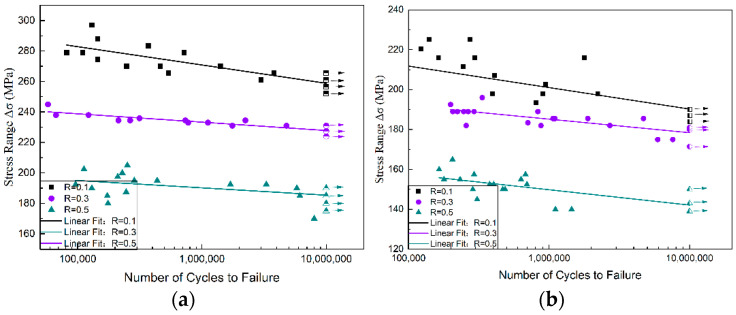
Fatigue S–N curves of the BM and FSW joints: (**a**) fatigue S–N curves of the BM subjected to R = 0.1, 0.3, and 0.5; (**b**) fatigue S–N curves of the FSW joints subjected to R = 0.1, 0.3, and 0.5.

**Figure 5 materials-15-08010-f005:**
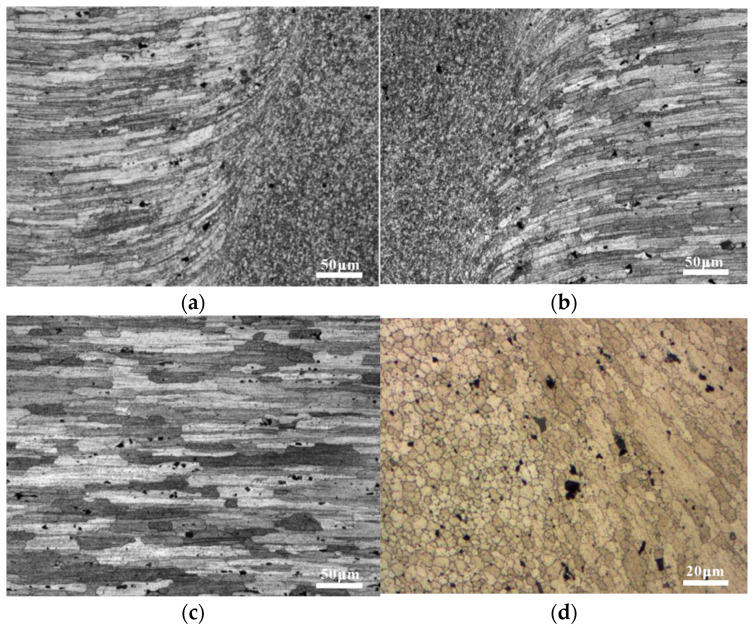
Microstructure of the 7050 aluminum alloy’s FSW joints: (**a**) the TMAZ-SZ; (**b**) the HAZ; (**c**) high-magnification morphology of the TMAZ-SZ; (**d**) secondary-phase morphology of the HAZ; (**e**) secondary-phase morphology of the SZ; (**f**) SEM of the BM.

**Figure 6 materials-15-08010-f006:**
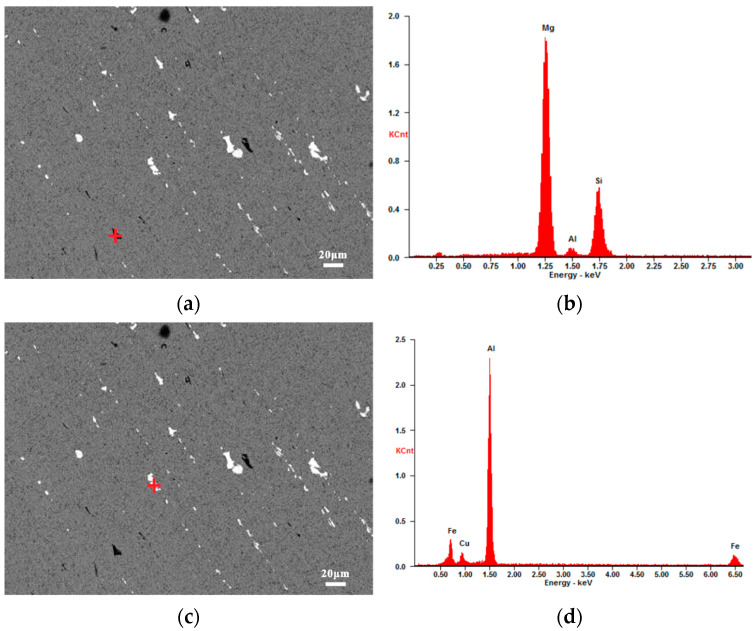
EDS spectra of the secondary phases: (**a**,**c**,**e**,**g**) SEM images of the matrix; (**b**) Mg_2_Si; (**d**) Al_23_CuFe_4_; (**f**) Al_2_Mg_3_Zn_3_; (**h**) Al_7_Cu_2_Fe.

**Figure 7 materials-15-08010-f007:**
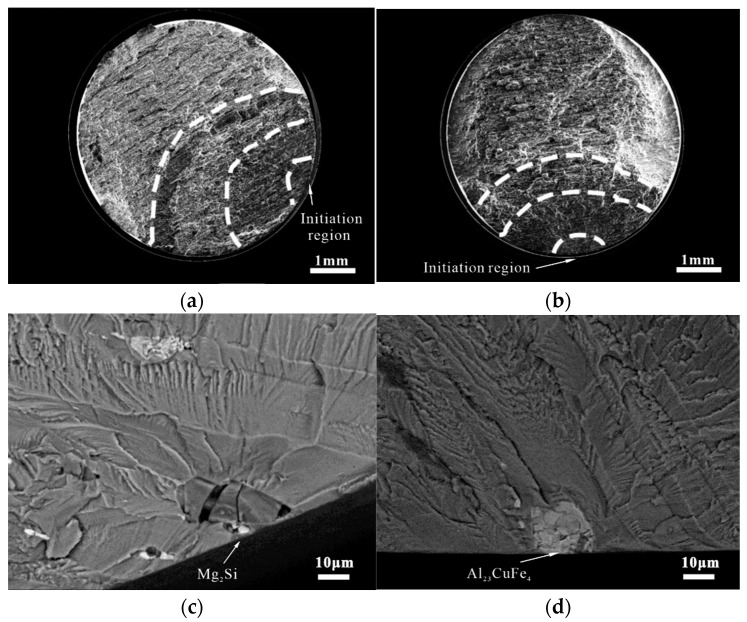
Fatigue fracture morphology: (**a**) M-3-1# sample macroscopic fracture; (**b**) J-3-23# sample macroscopic fracture; (**c**) M-3-1# sample origin morphology; (**d**) J-3-23# sample origin morphology.

**Figure 8 materials-15-08010-f008:**
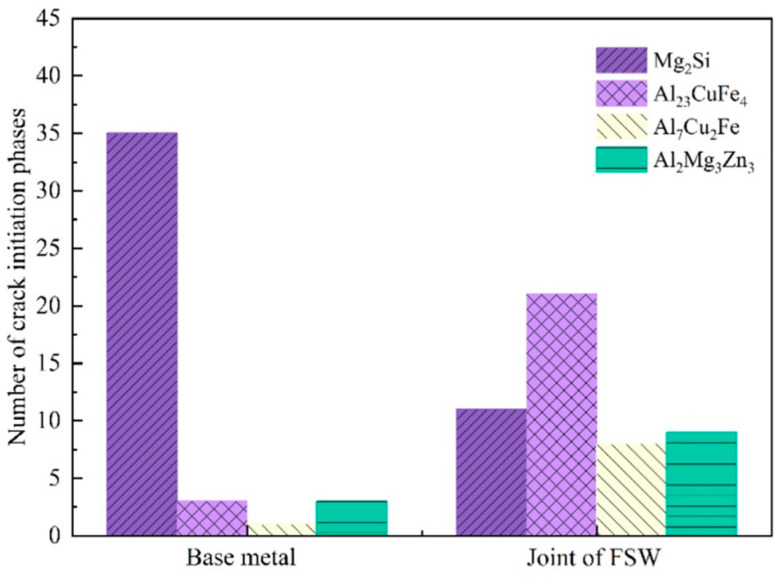
Initiation phase statistics for the BM and FSW joints.

**Figure 9 materials-15-08010-f009:**
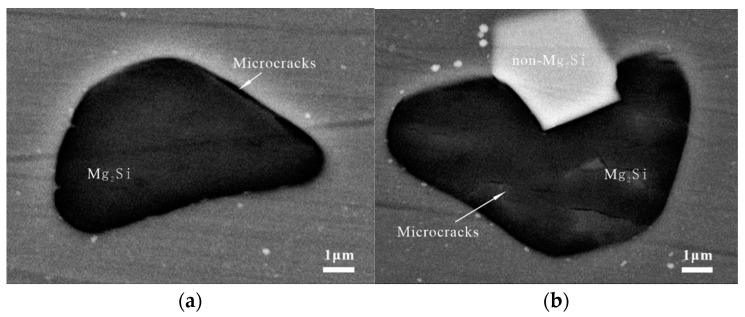
Coarse secondary phase in the initial state of the BM: (**a**) microcracks in the BM of the Mg_2_Si phase; (**b**) Mg_2_Si phase with internal microcracks and various non-Mg_2_Si phases.

**Figure 10 materials-15-08010-f010:**
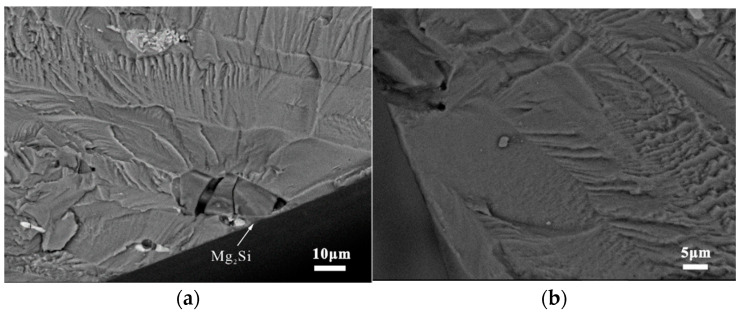
Mg_2_Si phase cracking at the BM specimens’ fracture: (**a**) Mg_2_Si phase internal fracture; (**b**) Mg_2_Si phase debonding, with serrated cracks near the edge.

**Figure 11 materials-15-08010-f011:**
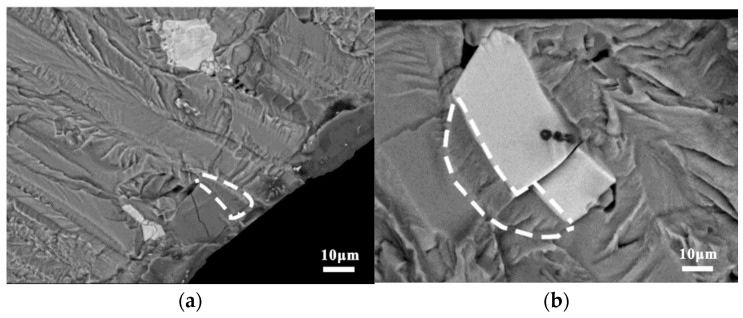
Interfacial type cracking in the BM specimens: (**a**) Mg_2_Si phase and (**b**) non-Mg_2_Si phase.

**Figure 12 materials-15-08010-f012:**
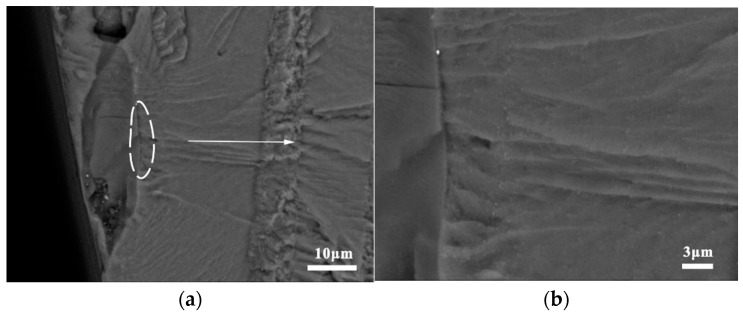
Cracking of the Mg_2_Si phase in joint specimens: (**a**) debonding of the Mg_2_Si phase and (**b**) serrated cracks.

**Figure 13 materials-15-08010-f013:**
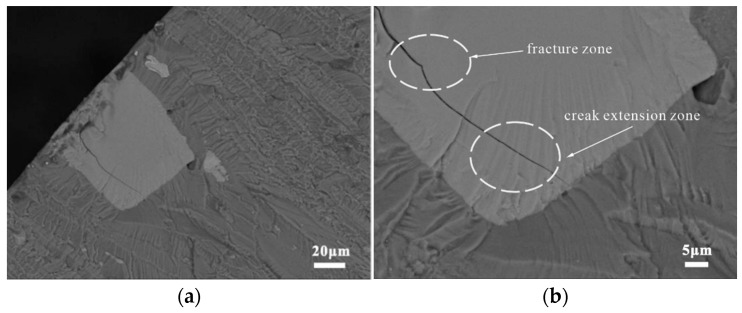
Cracking of the non-Mg_2_Si phase in joint specimens: (**a**) origin of the fracture pattern and (**b**) local magnification.

**Figure 14 materials-15-08010-f014:**
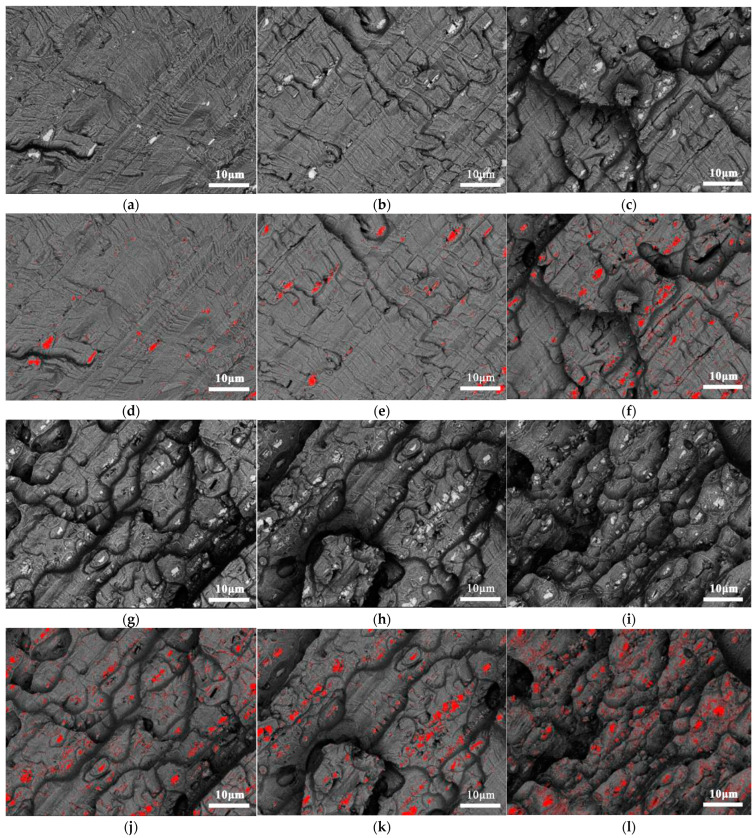
Distribution of the coarse secondary phase, during different stages of the fatigue crack extension: (**a**) early expansion stage; (**b**) stable expansion stage; (**c**) boundary between the stable and unstable expansion stages; (**g**) unstable expansion stage I; (**h**) unstable expansion stage II; (**i**) unstable expansion stage III. (**d**) processed image, (**e**) processed image, (**f**) processed image, (**j**) processed image, (**k**) processed image, (**l**) processed image.

**Figure 15 materials-15-08010-f015:**
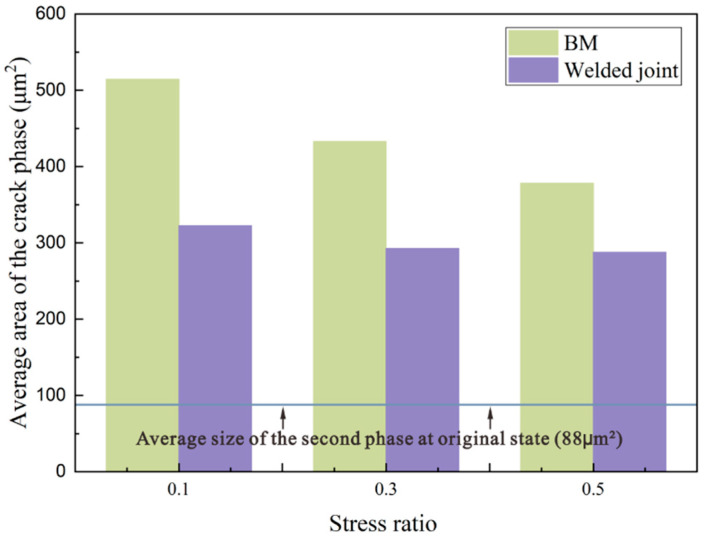
Average area of cracked phase for different stress ratios.

**Figure 16 materials-15-08010-f016:**
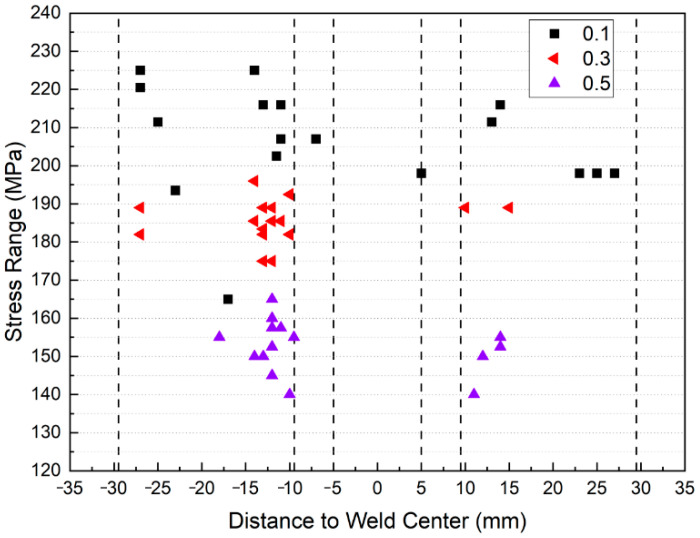
Characterization of the FSW joint fracture locations for different stress ratios.

**Table 1 materials-15-08010-t001:** Chemical composition of the 7050-T7451 aluminum alloy.

Zn	Mg	Cu	Fe	Si	Mn	Ti	Al
5.89	2.59	1.98	0.29	0.10	0.10	0.05	Remaining

**Table 2 materials-15-08010-t002:** Mechanical properties of the 7050-T7451 aluminum alloy.

R*_eL_*/MPa	R*_m_*/MPa	*A* (%)	HV
492	560	12	150

**Table 3 materials-15-08010-t003:** FSW parameters.

Spinning Speed /r·min^−1^	Welding Speed /mm·min^−1^	Inclination /°	Depression /mm
300	80	2.5	0.1

**Table 4 materials-15-08010-t004:** Summary statistics for different secondary phases.

Crack Initiation Phase	BM Specimen			FSW Specimen		
	R = 0.1	R = 0.3	R = 0.5	R = 0.1	R = 0.3	R = 0.5
Mg_2_Si	13	12	10	5	5	1
Al_23_CuFe_4_	0	0	3	6	6	9
Al_7_Cu_2_Fe	0	1	0	3	3	2
Al_2_Mg_3_Zn_3_	0	1	2	3	2	4

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
