# Peer review of "Fatigue Characteristics of 7050-T7451 Aluminum Alloy Friction Stir Welding Joints and the Stress Ratio Effect"

_materials, 2022, doi:10.3390/ma15228010_

Round 1

Reviewer 1 Report

1. Introduction: In the introduction section, the author must clearly express the research motivation, research contributions, and research gap. In fact, I don't see anything in the introduction part that has to do with these three essential things. 

2. Why and how can you recognise the parameters listed in Table 3? 

3. The experiment's weak design makes the results less dependable, hence a stronger design of the experiment needs to be carried out.

4.4.The article lacks discussion of the results in comparison to other articles as well as implications for academic and commercial matters.

Reviewer 2 Report

The paper is about the fatigue characteristics of 7050-T7451 aluminum alloy friction stir welding joints and the stress ratio effect. The paper is appropriate  for publication after considering the following comments:

1.  The scale bar should be added to SEM images.

2. What is the effect of fine precipitates on fatigue properties?

3. According to the following paper welding can affect the grain boundary types. What is the effect of grain boundaries on your study?

https://doi.org/10.1016/j.ijhydene.2022.04.260

4. In the SEM images, the area examined by EDS should be marked.

Reviewer 3 Report

This article focuses on the stress ratio effect on the fatigue characteristics of 7050-T7451 aluminum alloy friction stir welded joints. The methodology is well known and incorporated in many research papers in the same area such as

·       Effects of the heterogeneous microstructure of a 7050-T7451 aluminium alloy FSW joint on fatigue behaviour under different stress ratios

·       https://doi.org/10.1016/j.msea.2019.138223

·       http://dx.doi.org/10.1016/j.ijfatigue.2015.10.001

In nutshell, I do not find any novelty, uniqueness and innovative finding in the present research.

Round 2

Reviewer 1 Report

well done

Reviewer 2 Report

The paper can be accepted.

Reviewer 3 Report

The manuscript can be accepted in its present form.